



# High resolution bed level change and synchronized biophysical data from 10 tidal flats in northwestern Europe

Zhan Hu[1,2], Pim W.J.M. Willemsen[3,4], Bas W. Borsje[3], Chen Wang[5], Heng Wang[2,6], Daphne van der Wal[4,7], Zhenchang Zhu[8], Bas Oteman[4], Vincent Vuik[9,10], Ben Evans[11], Iris Möller[12], Jean-Philippe Belliard[13], Alexander Van Braeckel[14], Stijn Temmerman[13], Tjeerd J. Bouma[4,15]

[1]Guangdong Provincial Key Laboratory of Marine Resources and Coastal Engineering, and School of Marine Science, Sun Yat-sen University, Guangzhou, 510275, China
[2]Southern Marine Science and Engineering Guangdong Laboratory (Zhuhai), China
[3]Water Engineering and Management, University of Twente, Enschede, P.O. Box 217, 7500 AE, the Netherlands
[4]NIOZ Royal Netherlands Institute for Sea Research, Department of Estuarine and Delta Systems and Utrecht University, Yerseke, P.O. Box 140, 4400 AC, the Netherlands
[5]Satellite Application Center for Ecology and Environment, Ministry of Ecology and Environment, and State Environmental Protection Key Laboratory of Satellite Remote Sensing, Beijing, 100094, China.
[6]School of Marine Engineering and Technology, Sun Yat-Sen University, Guangzhou, 510275, China
[7]Faculty of Geo-Information Science and Earth Observation (ITC), University of Twente, Enschede, P.O. Box 217, 7500 AE, the Netherlands
[8]Institute of Environmental and Ecological engineering, Guangdong University of Technology, 510030 Guangzhou, China
[9]Delft University of Technology, Faculty of Civil Engineering and Geosciences, P.O. Box 5048, 2600 GA, Delft, the Netherlands
[10]HKV Consultants, P.O. Box 2120, 8203 AC, Lelystad, the Netherlands
[11]Department of Geography, University of Cambridge, Cambridge, CB2 3EN, UK
[12]Department of Geography, Trinity College Dublin, Dublin 2, D02 PN40, Ireland
[13]University of Antwerp, Ecosystem Management Research Group, Antwerp, B-2610 Belgium
[14]Research Institute for Nature and Forest (INBO), Havenlaan 88, 1000 Brussels, Belgium
[15]Department of Physical Geography, Utrecht University, Utrecht, P.O. Box 80.115, 3508 TC, the Netherlands

*Correspondence to*: Heng Wang (wangheng3@mail.sysu.edu.cn)

**Abstract.** Tidal flats provide valuable ecosystem services such as flood protection and carbon sequestration. Erosion and accretion processes govern the eco-geomorphic evolution of intertidal ecosystems (marshes and bare flats), and hence substantially affect their valuable ecosystem services. To understand the intertidal ecosystem development, high-frequency bed-level change data are thus needed. However, such datasets are scarce due to the lack of suitable methods that do not involve excessive labour and/or instrument cost. By applying newly-developed Surface Elevation Dynamics sensors (SED-sensors), we obtained unique high-resolution daily bed-level change data sets in the period 2013-2017 from 10 salt marsh sites situated in the Netherlands, Belgium and Britain in contrasting physical and biological settings. At each site, multiple sensors were deployed for 9-20 months to ensure sufficient spatial and temporal coverage of highly variable bed level change processes. The bed level change data are provided with synchronized hydrodynamic data, i.e. water level, wave height, tidal current velocity, and medium grain size (D50) as well as (for some sites) chlorophyll-a level and organic matter content of the surface sediment. This dataset has revealed diverse spatial morphodynamic patterns over daily to seasonal scales, which are valuable to theoretical and model development. On the daily scale, this dataset is particularly instructive as it includes a number of





storm events, the response to which can be detected in the bed level change observations. Such data are rare but useful to study
tidal flat response to highly energetic conditions.

The dataset is available from the 4TU.Centre for Research Data (https://doi.org/10.4121/uuid:4830dbc2-84b8-46f9-99a3-90f01ab5b923, Hu et al., 2020), which is expected to expand with additional SED-sensor data from ongoing and planned surveys.

**1 Introduction**

Salt marshes and the adjacent tidal flats are co-evolving coastal ecosystems with global importance (Mcowen et al., 2017; Schuerch et al., 2018). They provide multiple ecosystem services such as carbon sequestration (Mcleod et al., 2011; Duarte et al., 2013), hosting migratory birds (Van Eerden et al., 2005), and protecting coastal communities and infrastructures by attenuating waves (Temmerman et al., 2013; Möller et al., 2014; Vuik et al., 2016). These systems are known as dynamic
biogeomorphic systems (Knox, 1972; Friedrichs, 2011; Fagherazzi et al., 2012). Their bed form is continuously shaped by the interactions between physical and biological processes, including tidal currents, wind waves, sediment delivery, as well as bioturbation/bioaggregation, which jointly determine the time evolution of these systems (Le Hir et al., 2000; Yang et al., 2008; Green and Coco, 2014; Dai et al., 2016, 2018; D'Alpaos et al., 2016). The evaluation of these valuable coastal ecosystems with changing sea level and storminess is an issue that is of high socioeconomic concerns (Mariotti and Fagherazzi,
2010; Temmerman and Kirwan, 2015; Schuerch et al., 2018). More researches are clearly needed to reveal the key biogeomorphic processes that control the persistence of these intertidal ecosystems to enable an accurate assessment of their resilience.

Recent studies have shown that short-term (daily to seasonal scale) hydrodynamic forcing and the related bed level changes
exert a critical control on: i) the recruitment of marsh seedlings (Balke et al., 2014; Silinski et al., 2016; Cao et al., 2018) and benthic invertebrates (Bouma et al., 2001; Nambu et al., 2012); ii) initiation of marsh lateral erosion (Bouma et al., 2016) and iii) position and dynamics of the existing marsh edge (Willemsen et al., 2018; Evans et al., 2019). Large spatial (e.g. dense vegetation vs. bare) and temporal (e.g. stormy vs. calm) variation in bed level changes has been observed in intertidal systems (Spencer et al., 2016; Hu et al., 2017). Thus, to better understand intertidal bed level change and their impact on biogeomorphic
evolution, bed level change data with high resolution and sufficient spatio-temporal coverage are needed. However, such data are scarce to support theory and model development. For instance, we are lacking the ability to model cyclic marsh expansion-retreat dynamics since the existing data is insufficient to derive tipping points that lead to the expansion-retreat phase shift. Existing measurements of intertidal bed level dynamics typically have limited temporal (e.g., 2–5 tidal cycles) or spatial resolution (e.g., 1–2 stations) (Whitehouse and Mitchener, 1998; Shi et al., 2014; Zhu et al., 2014; Hunt et al., 2016), as high-
resolution data sets require excessive labour or high cost for instruments (Andersen et al., 2006).





In light of these limitations, SED-sensors (Surface Elevation Dynamics sensors) have been developed to record daily bed level dynamics with high accuracy, while reducing the unit cost and deployment labour (Hu et al., 2015). These sensors have been applied in the field at 10 sites in the Netherlands (Westerschelde and Wadden Sea), Belgium (Zeeschelde) and Britain (Thames and Humber Estuary) from a number of previous studies (Hu et al., 2017; Willemsen et al., 2018; Belliard et al., 2019). This paper presents a comprehensive collection of the existing SED-sensor dataset. It is expected to provide a rare opportunity to assist future studies on intertidal biogeomorphic processes as it offers: i) high temporal resolution (daily) bed level changes; ii) long temporal coverage, i.e. 9-20 months depending on the site; iii) large spatial coverage, i.e. multiple sensors deployed in both marshes and bare tidal flats across 10 sites; iv) synchronized biophysical measurements, i.e. hydrodynamic measurements (water level, flow velocity and significant wave height), sediment properties (grain size, chlorophyll-a level and organic matter content) and bathymetric/topographic profiles. In this paper, we present the full dataset from 10 sites, and briefly discuss the potential research questions that can be addressed by exploring the current dataset.

## 2 Site description

The current dataset includes 10 observation sites from the northwestern Europe: 7 sites from the Netherlands, 1 site from Belgium and 2 sites from Britain (Figure 1). For all the 7 Dutch sites, site 1-6 are in the Westerschelde estuary, and only site 7 is in the Wadden sea region. Near Zuidgors in the Westerschelde, there are two sites (sites 1 and 2). At site 1 (Zuidgors A), only the bare tidal flat was monitored, whereas at site 2 (Zuidgors B), both the bare tidal flat and marsh area were included in the monitoring. The only Belgian site (site 8 Galgeschoor) is located in the Zeeschelde estuary, which is the upstream part of the Westerschelde estuary. Site 8 has two observational transects: north and south transects with different bathymetries. The two British sites, site 9 (Tillingham) and 10 (Donna Nook) are on the southeast coast of England (Figure 1).

Overall, these 10 sites cover areas of differing tidal range, wave exposure, sediment grain size and marsh vegetation species (Table 1). Notably, site 10 (Donna Nook) has the largest tidal range (6.9 m), whereas sites 9 (Tillingham) have highest wave exposure. The observations were conducted in 2013-2017. The duration of the observation at each site varies from 9-20 months (Table 1). At all the sites, bed level changes were monitored daily with multiple SED-sensors. For all sites except site 1, 4 and 8, SED-sensors were deployed on both bare flat and marsh areas. The coordinates of the measuring stations as well as the bathymetry of the measuring transects were measured by Real Time Kinematic Global Positioning Systems (RTK-GPS) to an accuracy of 15 mm in the vertical and 10 mm in the horizontal. Besides the daily bed level observation, biophysical measurements were available at some sites, i.e. water level, wave height, current velocity, surface sediment grain size, chlorophyll-a level as well as organic matter content.



## 3. Method

### 3.1 bed level change observation

The bed level dynamics at each site were monitored using recently developed SED-sensors (Hu et al., 2015, see Figure 2). These sensors are standalone instruments with all the parts of measuring, data-logging and batteries enclosed in a transparent tube. The measuring part is an array of light sensitive cells that measure light intensity. When in use, a sensor is inserted vertically into the bed, leaving about half of the array above the bed. The cells above/below the bed receive different amount of the day light, which will lead to different voltage outputs in the array of cells. By using an autonomous script, the noise in the raw signal is reduced, and the bed level is determined as where the large transition from high to low voltage occurred (Figure. 2d, and see Willemsen et al., 2018). When bed accretion or erosion occur, the transition point moves up or down in the measuring array. Thus, by recording the changes of the transition point, we can measure the bed level changes. In some cases, scouring holes occurred around some of the deployed SED-sensors, with the maximum depth of 5 cm. They typically result in two transition points in the array, corresponding to the bottom and the top of the scouring holes. In such cases, bed level was determined as the vertical position at the top of the scouring holes. Details of SED-sensor data processing are included in Willemsen et al. (2018).

As the sensor is dependent on the presence of daylight, the measuring window is day time during low tide. Data acquired while the sensors were submerged or during night were excluded from the analysis. For most of the time, SED sensors provided at least one measurement per day, i.e. daily temporal resolution. To avoid recording bed level data when sensors were submerged, an effective measuring window was set as two hours around low tide. In such a window, we used the averaged readings as a bed level observation point.

The accuracy of the sensors has been compared to a precise manual method (i.e. Sedimentation Erosion Bar) (Hu et al., 2015). The manual measurements were conducted weekly from 13 June to 17 July 2014 at the second most seaward measuring station of site 1 (Zuidgors A). These observations serve as an independent quality control of our automatic SED-sensor measurements. Good agreement ($R^2$ = 0.89) has been obtained between these two methods (detailed in Hu et al., 2015). The estimated operational accuracy of the SED-sensors is 5.0 mm with a 3.9 mm standard deviation. Additionally, good agreement between the SED-sensors and Sedimentation Erosion Bar measurements has been obtained at site 8 (Galgeschoor) over an 18-months parallel measurement (Belliard et al., 2019).

### 3.2 Hydrodynamics measurements

Bed level changes in the intertidal environment are closely related to the local hydrodynamic forcing. We measured hydrodynamic parameters of water level, wave height and tidal current velocity simultaneously with the bed level measurement at some of our observation sites (Table 1). To measure the water level and wave height, we deployed pressure sensors 0.05 m-



0.10 m above the bed in the vicinity of the SED-sensors at some of the sites (see Table 1). For sites 1, 3, 4 and 6, OSSI-010-003C pressure sensors (Ocean Sensor Systems, Inc.) were used to measured pressure at a frequency of 5 Hz over a period of 7 mins, with a 15 mins interval. The mean water level is determined by the mean pressure in an interval. Significant wave height (Hs) and peak wave period (Tp) were derived from the dynamic wave pressure signals (Tucker and Pitt, 2001). For sites 2, 5, 9, 10, PDCR 1830 pressure sensors were used. Pressure was recorded at 4Hz for 4096 readings (~17 minutes) around high tide slack water, as determined by an on-board algorithm on the datalogger (Möller et al., 1999). This typically results in one set of wave parameters per tide. For site 8, both OSSI-010-003C and PDCR 1830 pressure sensors were used. The measuring frequency was 16 Hz for the PDCR sensors and 20 Hz for the OSSI-010-003C sensors. More details on the sensor deployments at site 8 are included in Belliard et al., (2019).

At sites without pressure sensor measurements, the water level data were obtained by nearby tidal gauge stations operated by Rijkswaterstaat (Dutch department of waterways and public works) or the British Oceanographic Data Centre (BODC). These data were obtained from Terneuzen (for site 2 Zuidgors B and site 5 Paulina) and Eemshaven (for site 7 Uithuizen) with 10 mins interval. For site 9 (Tillingham) and site 10 (Donna Nook), water level data were obtained at stations Sheerness and Immingham with 15 mins interval. Tidal current velocity was measured by Acoustic Doppler current profilers (ADCPs, Nortek Aquadopp) with a 5 or 10 mins interval at sites 1, 3 and 8. Additionally, near-bed 3D current velocities were measured at site 8 using two acoustic Doppler velocimeters (ADV, Nortek vectors). All the obtained hydrodynamic data are included in the current dataset.

**3.3 Sediment grain size and Chlorophyll-a monitoring**

To determine the median bed sediment grain size (D50), surface sediment samples (upper 2-3 cm) were collected at most of the sites (see Table 1). D50 of these samples was measured by Malvern laser particle sizer. Chlorophyll-a level in the sediment is an indicator for diatom biomass. Diatoms act as bio-stabilizer on tidal flats by producing Extracellular Polymeric Substances (EPS), and as such can affect sediment bed-level dynamics (Underwood & Paterson 1993; Austen et al., 1999; Andersen et al., 2005). At sites 2, 5, 9 and 10, chlorophyll-a samples were collected from the upper 1 cm of the sediment using a small cut-off syringe. The processing procedures that were used to determine chlorophyll-a are described in Willemsen et al. (2018). Additionally, at site 8, organic matter content was determined for the upper 2 cm of surface sediment samples by Loss on Ignition – LOI.

**4 Data description**

**4.1 Daily bed level changes with storm events**

At our study sites, daily bed level observations were conducted for 9-20 months, which includes conditions with various hydrodynamic forcing. As an example, we show the daily bed level change and the accompanying wave height data at site 4





(Zimmerman) and site 6 (Hellegat) from Feb-2015 to May-2016 (Figure 3). Waves in front (5 m) of the marsh cliffs at the site
4 (Zimmerman) were generally smaller than at site 6 (Hellegat) (Figure 3b vs. 3d). Additionally, at both sites, there was a
strong reduction in wave height from the bare tidal flats into the marshes (Figure. 3a vs. 3b and 3c vs. 3d). We observed that
the bed level fluctuation was more apparent on the bare tidal flats than in the marshes. Over the whole observation period, the
bed level fluctuation on the bare tidal flat was in the order of 5 cm at both sites, whereas bed level in the marshes stayed stable
(site 6 station 1) or experienced mild accretion (site 4 station 1).

Notably, a number of storm events with high incident waves were captured during our measurements. During the two storm
events in Nov 2015, Hs (significant wave height) exceeded 0.6 m on the bare flat stations at both sites (Figure 3e and 3f),
whereas the mean Hs over the whole observation period was 0.1 cm and 0.08 cm at these two stations, respectively. During
the two storm events, sudden erosion of 2-3 cm occurred on the two bare flat stations. However, bed level changes at the two
marsh stations remained small (0.5-1.0 cm). Across the 10 study sites, the most severe short-term erosion was observed at site
1 (Zuidgors A) on 27 and 28 October 2013 during the St. Jude storm (Hu et al., 2015). In that event, severe bed level erosion
of 10.5 cm depth was captured by our SED-sensor on one of the bare flat stations at site 1 (data not shown).

## 4.2 Seasonal bed level changes and bio-physical changes

Our observations at most sites were longer than 12 months. Thus, seasonal bed level changes were captured in our dataset.
Examples of seasonal bed level changes at site 2 (Zuidgors B), site 4 (Zimmerman) and site 6 (Hellegat) show complex
spatiotemporal variations (Figure 4). Our data show that all the stations at these three sites have alternating erosion and
accretion seasons. There is no consistent seasonal erosion-accretion pattern for all the stations. Winter is a typical season of
bed erosion for stations on the bare flat, but not for the stations in marshes.

Spatially, bed level variations were generally smaller at the landward stations in the marshes, and increased towards the
seaward stations at all three sites. We further observed that the most seaward station at the site 4 (Zimmerman) experienced
net erosion over an annual timescale, whereas stations at the other two sites were in equilibrium, i.e. the degree of erosion was
comparable to accretion. Profile elevation data show that marsh cliffs were distinct at site 2 (Zuidgors B) and site 6 (Hellegat),
with the cliff height being 0.88 m and 0.35 m, respectively, whereas a cliff was absent at site 4 (Zimmerman) (Figure 4).
Notably, the magnitude of bed level changes reduced from bare flat stations to the stations on the marsh plateaus at sites with
marsh cliffs (sites 2 and 6), whereas there was no clear difference between the bare flat station and the neighbouring marsh
station at the site without cliff (site 4).

## 4.3 Surface sediment characteristics

At 6 sites with surface sediment grain size measurements, two sites (sites 4 and 6) in the Westerschelde had the largest median
sediment grain size (Figure 5). At these two sites, D50 of the surface sediment was in the range of 66.7-131.8 μm, which was





significantly coarser than the rest of the shown sites ($p=0 < 0.05$). Within each site, there was no apparent difference in D50 between the marsh and bare flat stations around the marsh edge (50 m seaward and landward to the marsh edge). However, there was a gentle trend of coarsening from the landward to the seaward stations on bare flats.

Chlorophyll-a levels in surface sediment, a proxy for the diatom biomass and their bio-stabilization effect, were also obtained at some of our observation sites (Table 1). The Chlorophyll-a levels at the site 2 (Zuidgors B) showed great temporal variability (Figure 6). For all the stations, the Chlorophyll-a levels were generally low in winter (January), but reached their maximum at the end of the spring (May). However, there was no clear spatial pattern in the Chlorophyll-a levels across different stations, as the marsh stations had similar levels compared to the bare flat stations.

## 5 Data availability and future observations

All data presented in this paper are available from the 4TU.Centre for Research Data (see Hu et al., 2020, https://doi.org/10.4121/uuid:4830dbc2-84b8-46f9-99a3-90f01ab5b923). The repository includes data as well as instructions in readme files. Additionally, we expect that the current repository will expand with additional SED-sensor data from ongoing as well as planned future observation programs including mangrove wetlands, e.g. ANCODE project
(https://www.noc.ac.uk/projects/ancode).

## 6 Conclusions

By applying the novel high-resolution SED-sensors, we were able to perform long-term (e.g. a few months to a few years) monitoring of the bed elevation changes at daily frequency. Our observations have been carried out at 10 sites in three countries in Western Europe for a long duration (9-20 months). To our knowledge, the current dataset is the most complete and
comprehensive to date on high-resolution (daily) intertidal bed-level changes.

The SED-sensor data has been proven to be useful in revealing the relations between hydrodynamic forcing and intertidal bed level dynamics (Hu et al., 2018; Belliard et al., 2019) and understanding the spatial variations in bed level dynamics from tidal flats to salt marshes (Willemsen et al., 2018; Baptist et al., 2019). The presented dataset may be of further use to the scientific
community for addressing several research questions: In particular, our dataset can be used to provide insights on storm impacts on intertidal morphology and post-storm recovery (Leonardi et al., 2018), as the dataset pinpoints a number of storm events with precise pre- and post-storm bed level observations, which are otherwise difficult to measure by discontinuous manual methods. Furthermore, our dataset can be used to better understand biogeomorphic interactions in intertidal environments, which are important for marsh persistence, e.g. the control of short-term bed level changes on marsh seedling establishment
(Bouma et al., 2016; Cao et al., 2018), and the influence of marsh vegetation on sediment deposition (Yang et al., 2008; Schwarz et al., 2015; D' Alpaos and Marani, 2016). Lastly, our dataset may support morphodynamic model developments. Due to the lack of desired data, existing intertidal morphological models rarely deal with daily morphological changes. The



presented dataset contains high-resolution data across 10 sites with various spatially (marsh vs. bare flat) and temporally (calm vs. stormy) varying conditions, which is valuable for model development and evaluation. In addition to process-based

morphodynamic models (e.g. Delft3d, Lesser et al., 2004), this dataset can be of special interest to data-driven models based on machine learning techniques. Recent developments of the latter have shown great potential in resolving complex coastal morphodynamics (see a recent review in Goldstein et al., 2019). Therefore, the present dataset is expected to advance our understanding and prediction of tidal flat evolution and resilience.

### Acknowledgements

The authors gratefully acknowledge financial support of the Joint Research Project of National Natural Science Foundation of China (No. 51761135022) – NWO (No. ALWSD.2016.026) – EPSRC (No. EP/R024537/1): Sustainable Deltas, and Project of National Natural Science Foundation of China (No. 51609269). The dataset of Zuidgors A and Baarland was obtained as a part of the STW-NWO project (Grant No. 07324). The dataset of Hellegat, was obtained as part of the NWO funded project BE-SAFE (Grant No. 850.13.011). The dataset of Zuidgors B, Paulina, Tillingham and Donna Nook was obtained as part of

the EU FP7 funded project FAST (Foreshore Assessment using Space Technology) (Grant No. 607131). B Evans and I Möller received support from the UK NERC RESIST project (Grant No. NE/R01082X/1) for input into the paper preparation/writing process. The dataset of Galgeschoor was obtained in a project funded by Antwerp Port Authority.

### Author contribution

ZH, DV and TB developed the SED-sensor. ZH, PW, BB, DV, ZZ, BO, VV, BE, IM, JB, AB, ST, TB collected the raw data.
PW and HW processed the data. ZH, CW and HW prepared the manuscript with contributions from all authors.

### Competing interests

The authors declare that they have no conflicts of interest.

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



**Table 1: An overview of the observation sites in the NL, Be and, UK**

| Country | Site name/ estuary | Latitude/longitude | SED sensor time period | D50 mean and [spatial variations] (µm) | Tidal range (m) | Significant wave height mean and [standard deviation] (cm) | SED-sensor deployments relative to the marsh edge (m)[a] | Vegetation species | Bio-physical measurements[b] |
|---|---|---|---|---|---|---|---|---|---|
| NL | 1. Zuidgors A/ Westerschelde | 51°23'15.61"N, 3°49'43.46"E | 2013.10-2015.1 | 72.1 [23.4-202.1] | 4.3 | 8 [8] | 15, 64, 109, 150, 233, 308, 329, 346, 379 | *Spartina anglica, Salicornia spp* | D50, Hs, WL, Vel |
| | 2. Zuidgors B/ Westerschelde | 51°23'21.95"N, 3°50'7.51"E | 2015.9-2016.9 | [23.4-48.8] | 4.3 | 8 [8] | -20, -.5, 5, 25, 60, 100, 155 | *Spartina anglica, Salicornia spp* | D50, chl-a |
| | 3. Baarland/ Westerschelde | 51°23'49.56"N, 3°52'51.63"E | 2013.10-2015.1 | 26.8 [12.9-49.4] | 4.1 | 1 [1] | 12, 29, 38 | *Spartina anglica, Salicornia spp* | D50, Hs, WL, Vel |
| | 4. Zimmerman/ Westerschelde | 51°24'8.05"N, 4°10'32.15"E | 2015.1-2016.5 | [66.7-99.5] | 4.9 | 10 [7] | -50, -15, -5, 5 | *Spartina anglica, Salicornia spp* | D50, Hs, WL |
| | 5. Paulina/ Westerschelde | 51°20'59.73"N, 3°43'3.37" | 2014.12-2015.8 | [27-42.4] | 4.1 | 5 [3][c] | -42.5, -25.5, -17.5, -2.5, 22.5, 47.5, 127.5 | *Spartina anglica, Salicornia spp* | D50, chl-a |
| | 6. Hellegat/ Westerschelde | 51°21'59.33"N, 3°56'44.67"E | 2015.1-2016.5 | [113.4-131.8] | 4.2 | 11 [8] | -50, -15, -5, 5 | *Spartina anglica, Salicornia spp* | D50, Hs, WL |
| | 7.Uithuizen/ Wadden Sea | 53°27'24.57"N, 6°39'32.07"E | 2015.3-2016.4 | 87[d] | 4.0 | 7 [8] | -15, -10, -5, 2.5 | *Salicornia europaea, Puccinellia maritima, Spartina anglica* | WL |
| BE | 8. Galgeschoor/ Zeeschelde | 51°19'6.41″ N, 4°16'51.22″ E (North transect) 51°18'32.21″ N, 4°16'54.82″ E (South transect) | 2015.10-2017.5 | [28.6-259.1] | 5.2 | 7 [2] | 10, 150 (North transect) 9, 135 (South transect) | *Phragmites australis* | D50, Hs, WL, Vel |
| UK | 9. Tillingman/ Thames | 51°41'40.37"N, 0°56'32.80"E | 2015.7-2016.7 | 22.0 [5.5-70.3] | 4.8 | 17 [8] | -5, 7.5, 40, 52.5, 125, 130 | *Puccinellia maritima, Spartina anglica, Salicornia europaea* | D50, WL, chl-a, Hs |
| | 10. Donna Nook/ Humber | 53°29'28.20"N 0° 6'56.85"E | 2015.1-2015.10 | 153.8 [33.1-258.2 ] | 6.9 | 6 [5] | -2.5, 17.5, 35, 40, 45, 50 | *Puccinellia maritima, Spartina anglica,, Atriplex portulacoides* | D50, WL, chl-a, Hs |

[a] Positive/ negative values mean the deploy locations are in the seaward/ landward direction of the marsh edges. The exact GPS coordinates of the SED sensor deployment are included in the data file.

[b] Bio-physical measurements include: water level (WL), significant wave height (Hs), tidal current velocity (Vel), medium grain size (D50) and chlorophyll-a level (chl-a) of the surface sediment.

[c] This data is from Callaghan et al., (2010).

[d] This data is from Folmer et al., (2017).

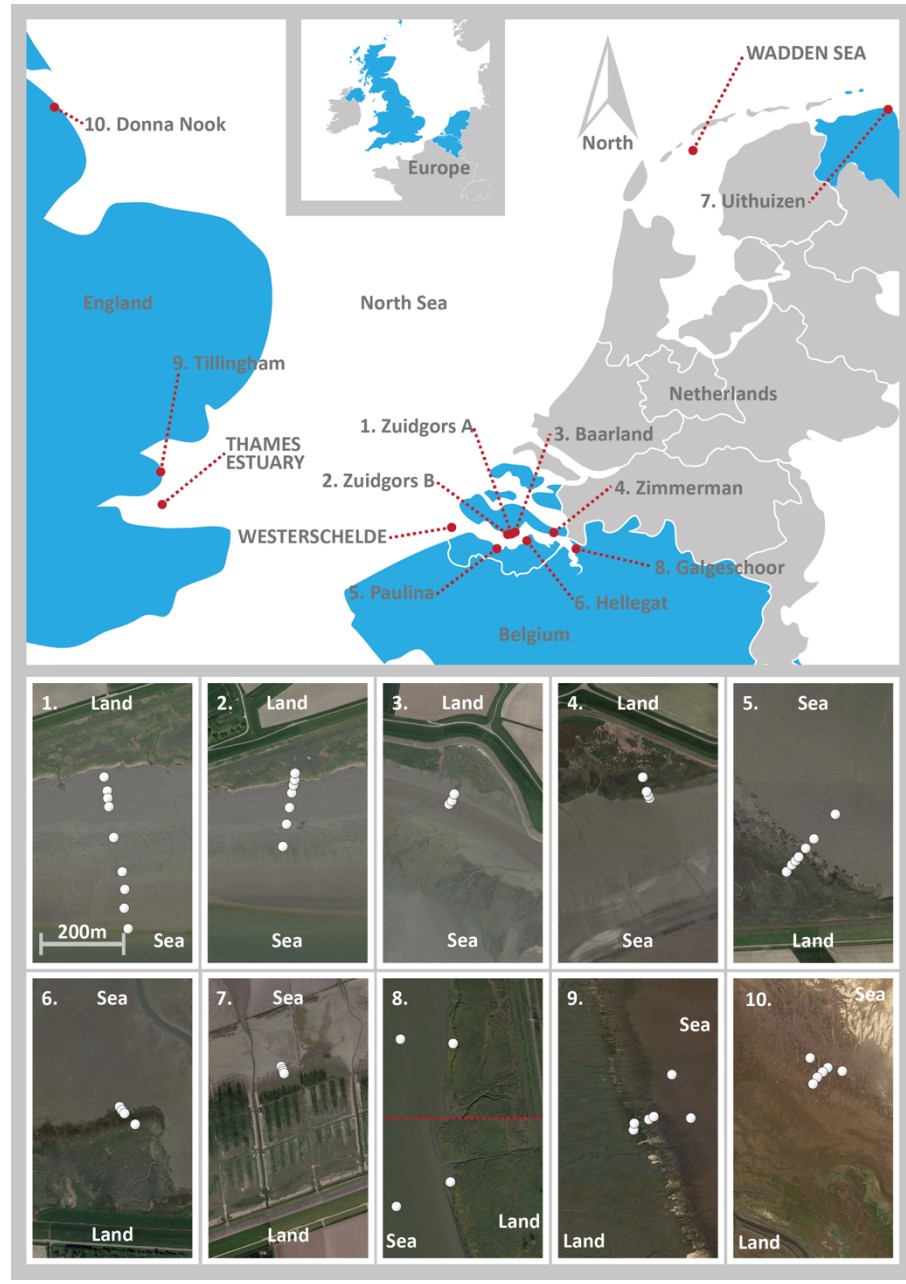


**Figure 1. Locations and an overview of the observation sites in the Westerschelde and Wadden Sea in the Netherlands, Zeeschelde in Belgium, Thames Estuary and Humber Estuary in Britain. The lower 10 panels are the deployments of the SED-sensor stations (white dots) at the observation sites. The GPS coordinates of all the SED-sensor stations are included in the data file. In panel 8, the maps of the north and south measuring transects at the site 8 (Galgeschoor) are bordered by a red dot line. Source of aerial imageries: ©Google earth.**




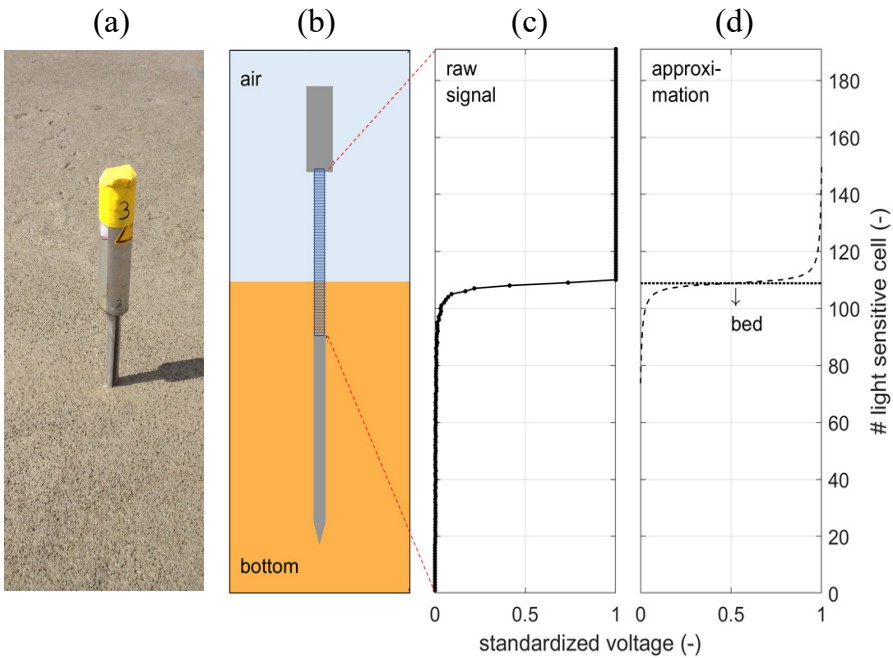

**Figure 2.** A photo (a) and schematization (b) of a SED-sensor in operation. The sensor uses an array of light-sensitive cells to determine the position of the bed level, resulting in a transection in the raw voltage output of the array (c). The noise in the raw signal is reduced, and the bed level is obtained by approximating the signal by an autonomous script (d). Details of SED-sensor data processing are included in Willemsen et al. (2018).



**Figure 3. Time series of bed-level change and significant wave height (Hs) at site 4 (Zimmerman, a, b, e, g) and site 6 (Hellegat, c, d, f, h). The top four planes (a-d) are the entire dataset of four measuring stations from these two sites. Stations 4s are on bare tidal flat (5 m seaward of the marsh edge), whereas station 1s are in the marsh (50 m landward of the marsh edge). The gaps in the bed 410 level time series were due to temporary sensor failures. The lower four planes are the enlarged plots of the stormy period in Nov 2015-Dec 2015. The dark grey shaded areas indicate bed-level changes during two storm events (13–17 Nov and 27 Nov–1 Dev in 2015).**

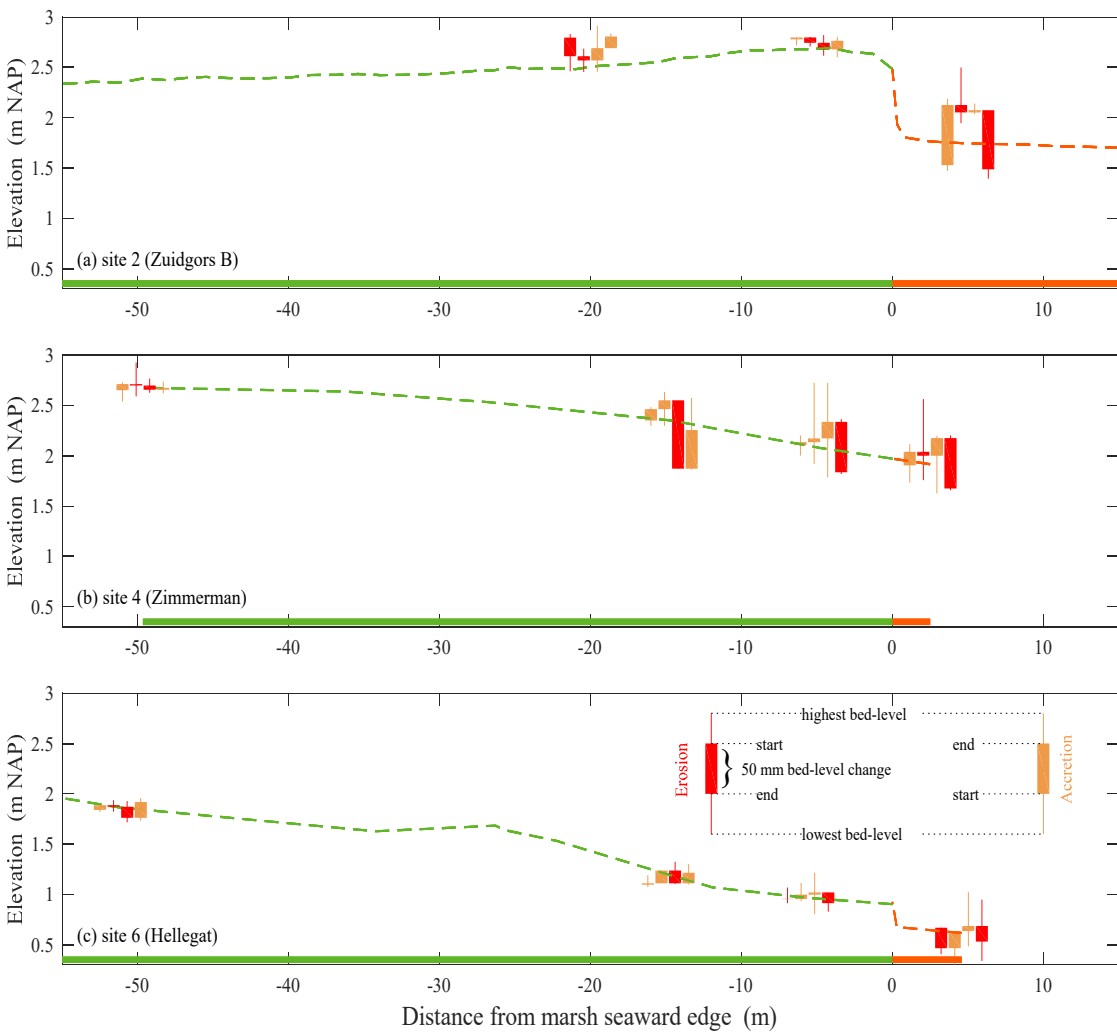

**Figure 4. Seasonal bed level changes at (a) site 2 (Zuidgors B), (b) site 4 (Zimmerman) and (c) site 6 (Hellegat) with bathymetry data (dash green line). At each SED-sensor station, the four bars from left to right indicate bed level changes in spring (Mar–May), summer (Jun–Aug), fall (Sep–Nov) and winter (Dec–Feb). The red bars indicate net erosion and the yellow bars indicate net accretion. The observed highest and lowest bed levels in a season are indicated by the high and low tips of the bars.**

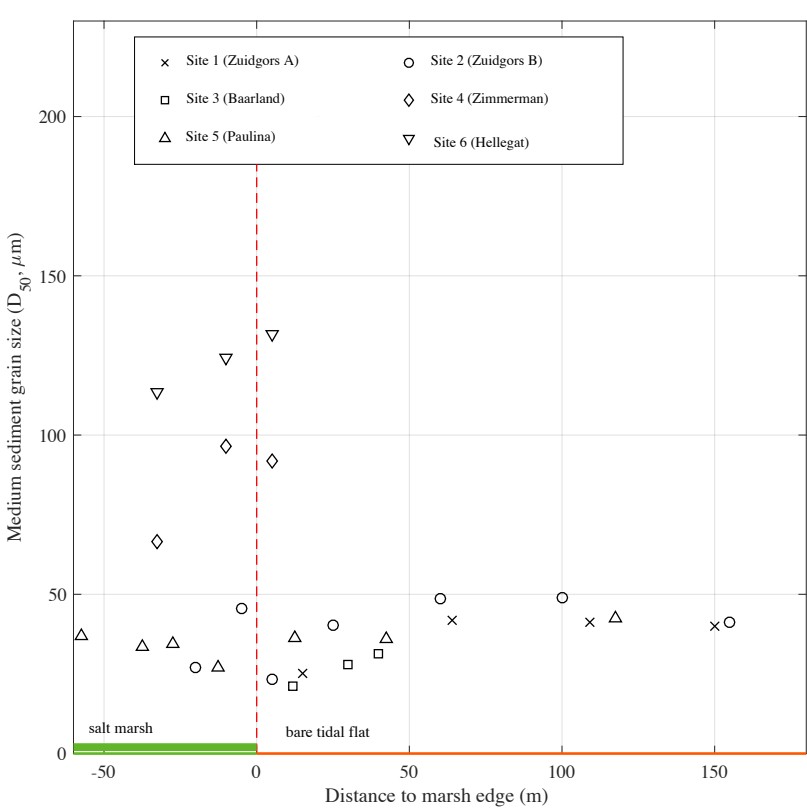

430 **Figure 5. Median grain size of surface sediment (D50) measured along cross-shore transects of six study sites in the Westerschelde estuary.**

435

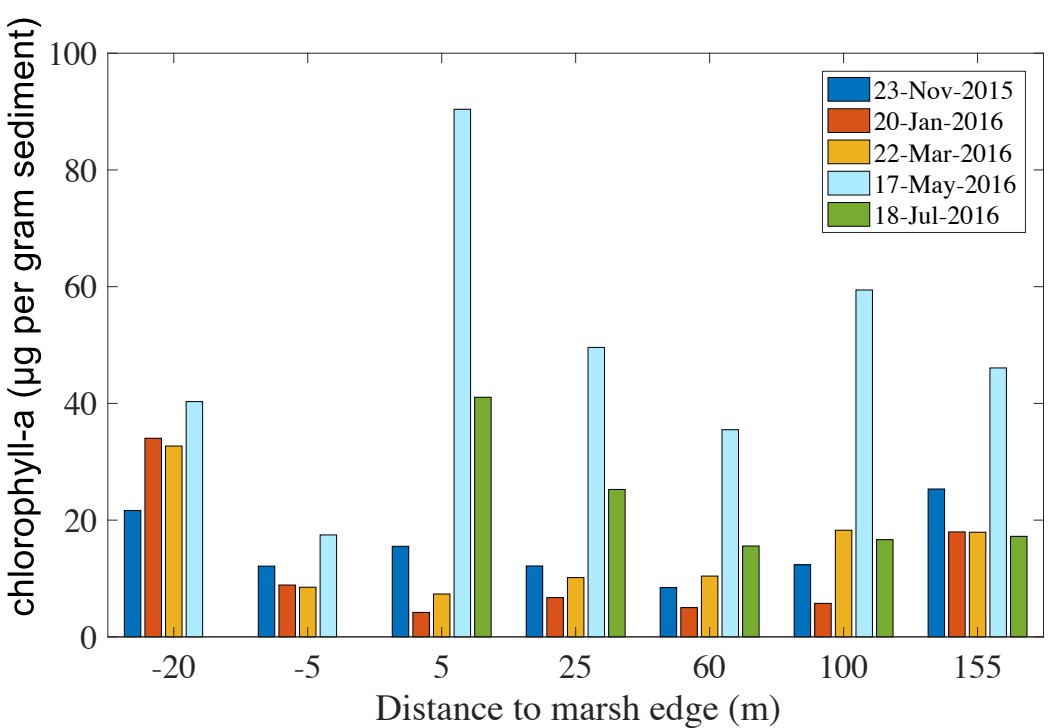

**Figure 6. Spatiotemporal variation of chlorophyll-a level in surface sediment (top 1 cm) at site 2 (Zuidgors B)**

440