# Peer review of "Synchronized high-resolution bed-level change and biophysical data from ten marsh-mudflat sites in northwestern Europe"

_Earth System Science Data, 2020_

## Referee Comment (RC1) · Edward Anthony (Referee) · 8 Jul 2020

This is an interesting and useful paper that opens up on bed-level changes and their drivers in tidal flats in NW Europe. The paper basically presents high-resolution bed-level data obtained from SED sensors, an innovative method of measuring short-term substrate variations on tidal flats, coupled with biophysical characteristics, and shows the potential link with drivers, notably storms. My main concern revolves around the wave-height measurements (data description and table 1).

Suggestions for improvement basically concern presentation problems listed below.

[Figure]

The term bed level needs to be hyphenated throughout when used in the adjectival form of 'bed-level changes' (this is done in Fig. 4c!). I would consider data as being in the plural form: 'data are' instead of 'data is', etc.

1. Title: Awkward title. High-resolution bed-level change is linguistically rather strange as I do not see how bed level change can be of high or low resolution. Measurements or data are of high or low resolution but not change. I suggest 'Synchronized high-resolution bed-level and biophysical data from .....'

2. Abstract: Line 73: deployment of labour Line 76: delete 'rare' to read: providing an opportunity

3. Site description: Line 84: delete 'the' to read from northwestern Europe and use lettering when indicating the number of sites: seven sites, one site, etc. Site 1, 7 etc is OK as you are identifying your sites. Line 96: repetition of measuring. Maybe use: The coordinates of the monitoring stations as well as the bathymetry of the measuring transects...

4. Methods: I know this is not a study on long-term net sedimentation-erosion rates but how do you account for subsidence from the SED data? Line 143: water-level data were obtained from nearby tidal gauge stations... Line 144: Diatoms act as bio-stabilizers...

5. Data description: Line 173: I am highly surprised by mean Hs values of 0.1 cm and 0.008 cm? These are meaningless and tantamount to zero wave energy! How did you measure such values? A 1 mm-high wave!!! I would suggest you refrain from quoting these values (this also goes for the very low Hs values in Table 1), and focus more on the higher Hs values and their variations in Figure 3 which are more meaningful and pertinent to bed-level change.

6. Conclusions Line 22: Due to the lack of 'relevant' (instead of desired).

7. Table and figures: Table 1: Surprised at the significant wave height data values!

Figure 1: Please add geographical coordinates and scale. Figure 2: Please add a scale in (b) to show length of SED probe. Figure 3. Use black bar to indicate wave height in figure legend.

[Figure]

---

## Referee Comment (RC2) · Alvise Finotello (Referee) · 20 Nov 2020

This paper presents a dataset containing high-resolution, short term measurements of bed-level changes and synchronized biophysical data (water level, water velocity, sediment grain-size, chl-a, wave height) in coupled tidal-flat and salt-marshes systems across northern Europe. Bed-level change data were retrieved from newly-developed SED sensors. These sensors were deployed for time-spans ranging between 9 to 20 months from 2013 to 2017.

The article is clear and well structured. Figures and tables are of good quality and appropriate to support the text. The dataset presented is most certainly new (even

though some subsets of these data were already employed for individual studies on specific study-case), and I believe it has a strong potential for being useful in the future (also because the authors intend to further expand the dataset by adding new study cases and data).

Methods and material are described in sufficient detail (with some minor exceptions that I've listed here below), and the cited literature is appropriate in my view.

I encountered no issues in accessing the dataset, which I found is complete and well organized. I've only highlighted in my comments one minor suggestion regarding the format of geographic coordinates to improve data consistency throughout the dataset. Data are easy to read and ready-to-use.

However, I share the concern of Referee 1 (see RC1 comments n.5 and n.7) regarding the reported low values of significant wave height (Hs) equal to 0.1 and 0.08 cm (text line 173). The authors refer to the work of Tucker and Pitt (2001) for the calculation of Hs and wave period from dynamic wave pressure signals. I believe the text should be briefly expanded to provide an explanation regarding this procedure, which are the potential sources of error, and what is the physical limit – if any – below which the derived Hs values should be disregarded due to (e.g.) instrument accuracy.

DETAILED COMMENTS

l.1: the wording chosen for the title is a bit strange, because it seems that "high-resolution" is referred to bed level (hyphenation is missing, by the way). I would suggest changing it to "High-resolution synchronized data of bed-level changes and biophysical data from. . .", or something similar. Reading the abstract it becomes clear that the focus of the paper is also (if not mostly) on salt marshes, so I find the choice of using "tidal flats" in the title a bit odd. My impression is that making explicitly clear that the paper aims to investigate the dynamics of both tidal flats and marshes would be much better.

l.53-54: "The evaluation of….", there's something strange in this sentence. Perhaps "Evaluating the impact of changing sea level and increasing storminess on these valuable coastal ecosystems is of socioeconomic importance" is more clear.

l.118-120: how was the low-tide defined/measured? Directly by the SED sensor (if so, how) or rather by coupling SEDs with a pressure gauge to measure water elevation? Later in the text (l.133) it is specified that "we deployed pressure sensors 0.05 m 0.10 m above the bed in the vicinity of the SED-sensors at some of the sites (see Table 1)". I think the use of pressure sensors should be mentioned in l.118-120 too.

l.105-115: it would be nice to add some information here (or in Fig.2) regarding the size (i.e., total length) of the SED sensors.

DATA: in the "profile_all_Mar_2020.xlsx" file, the coordinates (LAT, LON) for all sites are reported in decimal degrees except for sites 9 and 10. I believe it'd be better to keep the decimal degree format throughout the whole dataset. This should be an easy fix. Also, I've noticed that coordinates for site 10 are reported in red: is there a specific reason behind this color choice or is it just a misprint?

FIGURE: Figure 1: the use of blue color to highlight (some) land cover makes the figure a bit confusing I think. It would be probably better to lose such a color-code in favor of a much clearer one with gray=land and white=sea.

MINOR COMMENTS

l.31: "instrument cost" –> "costly instruments" is perhaps better.

l.33: "salt marsh sites"->"salt-marsh sites".

l.34-35: "bed level changes"->"bed-level changes" (please add hyphen throughout the whole manuscript).

l.36: D50->D_50 (use subscript throughout the whole manuscript)

l.36: "some sites", please specify how many.

l.46: "with"-> "of" global importance.

l.61: a semicolumn is missing after "(Bouma et al., 2016)".

l.82: "the current"-> "such"

l.84: lose "current"

l.94: "in 2013-2017" -> "in the period 2013-2017"; also "from 9-20 months"->"from 9 to 20 months"

---

## Author Comment (AC1) · 4 Dec 2020

Editor Earth System Science Data

Dec 4, 2020

Dear Editor,

We would like to thank you for your rapid handling and response to our submission. Please find enclosed the revised manuscript for the Earth System Science Data, entitled "Synchronized high-resolution bed-level change and biophysical data from ten marsh-mudflat sites in northwestern Europe" [Paper # essd-2020-78], and detailed list

of our responses to the comments of the two reviewers.

We highly appreciated the comments made by the associate editor and reviewers, as they enabled us to improve our manuscript. In the attached letter below, we have detailed how their comments were used to revise the manuscript. If you have further suggestions for improvement, we will be willing to make additional modification.

Looking forward to your reply.

Yours sincerely, On behalf of all co-authors, Heng Wang

E-mail: wangheng3@mail.sysu.edu.cn

"Synchronized high-resolution bed-level change and biophysical data from ten marshmudflat sites in northwestern Europe" [Paper # essd-2020-78] Zhan Hu, Pim W.J.M. Willemsen, Bas W. Borsje, Chen Wang, Heng Wang, Daphne van der Wal, Zhenchang Zhu, Bas Oteman, Vincent Vuik, Ben Evans, Iris Möller, Jean-Philippe Belliard, Alexander Van Braeckel, Stijn Temmerman, Tjeerd J. Bouma

Response to reviewers' comments 1. Response to Edward Anthony (Referee 1): 1.1 This is an interesting and useful paper that opens up on bed-level changes and their drivers in tidal flats in NW Europe. The paper basically presents high-resolution bed level data obtained from SED sensors, an innovative method of measuring short-term substrate variations on tidal flats, coupled with biophysical characteristics, and shows the potential link with drivers, notably storms. My main concern revolves around the wave-height measurements (data description and table 1).

Response: We would like to thank the positive comments from the reviewer. We hope that the revised version responds adequately to your concerns and we remain available for further improvement. The concern related to the wave-height measurements is addressed in detail in point 1.7 below.

1.2 The term bed level needs to be hyphenated throughout when used in the adjectival form of 'bed-level changes' (this is done in Fig. 4c!). I would consider data as being in

the plural form: 'data are' instead of 'data is', etc.

Response: We thank the reviewer for these linguistic suggestions. The suggested revisions are done for the whole manuscript. Data and dataset are both considered in the plural form.

1.3 Title: Awkward title. High-resolution bed-level change is linguistically rather strange as I do not see how bed level change can be of high or low resolution. Measurements or data are of high or low resolution but not change. I suggest 'Synchronized high-resolution bed-level and biophysical data from .....'

Response: We thank this suggestion. The new title of the manuscript is: "Synchronized high-resolution bed-level change and biophysical data from ten marsh-mudflat sites in northwestern Europe".

1.4 Abstract: Line 73: deployment of labour; Line 76: delete 'rare' to read: providing an opportunity

Response: We thank these suggestions from the reviewer and have revised these parts of the manuscript accordingly. "In light of these limitations, SED-sensors (Surface Elevation Dynamics sensors) have been developed to record daily bed-level dynamics with high accuracy, while reducing the unit cost and labour during deployment (Hu et al., 2015)" (Line 71)

1.5 Site description: Line 84: delete 'the' to read from northwestern Europe and use lettering when indicating the number of sites: seven sites, one site, etc. Site 1, 7 etc is OK as you are identifying your sites. Line 96: repetition of measuring. Maybe use: The coordinates of the monitoring stations as well as the bathymetry of the measuring transects...

Response: We thank these suggestions for the reviewer and have revised these parts of the manuscript accordingly.

1.6 Methods: I know this is not a study on long-term net sedimentation-erosion rates

СЗ

but how do you account for subsidence from the SED data? Line 143: water-level data were obtained from nearby tidal gauge stations... Line 144: Diatoms act as biostabilizers...

Response: We thank the reviewer for this comment. Indeed, our study is not for long-term net sedimentation-erosion rates, but focusing more on short-term bed-level changes. Thus, we did not account for subsidence. Over a short-term deployment (10-15 months), the SED sensor data has a good agreement with other manual measurements as shown in Hu et al., (2015) and Willemsen et al., (2018). Nonetheless, to account for subsidence, we can use regular RTK-GPS measurement of the bed surface as an independent data source for correction. We have now included this information in the revised manuscript in Line 113 as: "For a typical deployment period of the SED sensors (10-15 months), subsidence does not form a main source of error, as indicated by the good agreement between SED sensor data and other manual measurements (as shown in Hu et al., 2015 and Willemsen et al., 2018). Nonetheless, regular RTK-GPS measurements of bed level can serve as independent data sources for subsidence correction." The suggested revisions on the use of hyphen are adapted in the revised manuscript: 'water level data' to 'water-level data' and 'bio-stabilizers' to 'biostabilizers'.

1.7 Data description: Line 173: I am highly surprised by mean Hs values of 0.1 cm and 0.008 cm? These are meaningless and tantamount to zero wave energy! How did you measure such values? A 1 mm-high wave!!! I would suggest you refrain from quoting these values (this also goes for the very low Hs values in Table 1), and focus more on the higher Hs values and their variations in Figure 3 which are more meaningful and pertinent to bed-level change.

Response: We thank the reviewer for this comment. We apologize that the very low Hs values (0.1 cm and 0.008 cm) here were due to a mistake in the calculation. The corrected values should be 10.4 cm and 11.3 cm, respectively. We also checked the mean Hs values in the Table 1, which are correct. Similar values can be found in previous field studies at these sites (e.g. Callaghan et al., 2010; Hu et al., 2015). The very low mean Hs value (1 cm) at Site 3 Baarland in this table is because this site is sheltered by an offshore sandbar that blocks almost all the incident waves (see Figure 1). Additionally, we provide more details related to the data-processing method of Tucker and Pitt (2001) and the source of the processing routines in Line 138: "Significant wave height (Hs) and peak wave period (Tp) were derived from the dynamic wave pressure signals. The attenuation of pressure signals with water depth was corrected using the standard calculations methods as described in Tucker and Pitt, (2001). The attenuation correction was only applied over the frequency range 0.05-0.4 Hz and the maximum correction factor was set as 5 to avoid over-amplification of high frequency signals (i.e. noise). A detailed description and source of the data-processing routines can be found at http://neumeier.perso.ch/matlab/waves.html."

1.8 Conclusions Line 22: Due to the lack of 'relevant' (instead of desired).

Response: We thank the reviewer's suggestion. The text is revised accordingly.

1.9 Table and figures: Table 1: Surprised at the significant wave height data values! Figure 1: Please add geographical coordinates and scale. Figure 2: Please add a scale in (b) to show length of SED probe. Figure 3. Use black bar to indicate wave height in figure legend.

Response: We thank reviewer's comments on the table and figures. The issue with significant wave height is addressed in point 1.7. All the suggestions with the figures are adapted in the revised manuscript. Please see the updated figures below.

2. Response to Alvise Finotello (Referee 2) 2.1 The article is clear and well structured. Figures and tables are of good quality and appropriate to support the text. The dataset presented is most certainly new (even though some subsets of these data were already employed for individual studies on specific study-case), and I believe it has a strong potential for being useful in the future (also because the authors intend to further expand the dataset by adding new study cases and data). Methods and material are described

in sufficient detail (with some minor exceptions that I've listed here below), and the cited literature is appropriate in my view. I encountered no issues in accessing the dataset, which I found is complete and well organized. I've only highlighted in my comments one minor suggestion regarding the format of geographic coordinates to improve data consistency throughout the dataset. Data are easy to read and ready-to-use.

Response: We would like to thank the positive comments from the reviewer. We hope that the revised version responds adequately to your concerns and we remain available for further improvement.

2.2 However, I share the concern of Referee 1 (see RC1 comments n.5 and n.7) regarding the reported low values of significant wave height (Hs) equal to 0.1 and 0.08 cm (text line 173). The authors refer to the work of Tucker and Pitt (2001) for the calculation of Hs and wave period from dynamic wave pressure signals. I believe the text should be briefly expanded to provide an explanation regarding this procedure, which are the potential sources of error, and what is the physical limit – if any – below which the derived Hs values should be disregarded due to (e.g.) instrument accuracy.

Response: We thank the reviewer for this comment. This response was shared with that of the comment 1.7. We apologize that the very low Hs values (0.1 cm and 0.008 cm) were due to a mistake in the calculation. The corrected values should be 10.4 cm and 11.3 cm, respectively. We also checked the mean Hs values in the Table 1, which are correct. Similar values can be found in previous field studies at these sites (e.g. Callaghan et al., 2010; Hu et al., 2015). The very low mean Hs value (1 cm) at Site 3 Baarland is because this site is sheltered by an offshore sandbar that blocks almost all the incident waves (see Figure 1). Additionally, we provide more details related to the data-processing method of Tucker and Pitt (2001) and the source of the processing routines in Line 138:

"Significant wave height (Hs) and peak wave period (Tp) were derived from the dynamic wave pressure signals. The attenuation of pressure signals with water depth was corrected using the standard calculations methods as described in Tucker and Pitt, (2001). The attenuation correction was only applied over the frequency range 0.05-0.4 Hz and the maximum correction factor was set as 5 to avoid over-amplification of high frequency signals (i.e. noise). A detailed description and source of the data-processing routines can be found at http://neumeier.perso.ch/matlab/waves.html."

2.3 I.1: the wording chosen for the title is a bit strange, because it seems that "high resolution" is referred to bed level (hyphenation is missing, by the way). I would suggest changing it to "High-resolution synchronized data of bed-level changes and biophysical data from...", or something similar. Reading the abstract, it becomes clear that the focus of the paper is also (if not mostly) on salt marshes, so I find the choice of using "tidal flats" in the title a bit odd. My impression is that making explicitly clear that the paper aims to investigate the dynamics of both tidal flats and marshes would be much better.

Response: We would like to thank the reviewer for this constructive comment. Now the title is revised as: "Synchronized high-resolution bed-level change and biophysical data from ten marsh-mudflat sites in northwestern Europe"

2.4 I.53-54: "The evaluation of...", there's something strange in this sentence. Perhaps "Evaluating the impact of changing sea level and increasing storminess on these valuable coastal ecosystems is of socioeconomic importance" is more clear. Response: We agree with the reviewer's comment, and the text is revised accordingly.

2.5 I.118-120: how was the low-tide defined/measured? Directly by the SED sensor (if so, how) or rather by coupling SEDs with a pressure gauge to measure water elevation? Later in the text (I.133) it is specified that "we deployed pressure sensors 0.05 m 0.10 m above the bed in the vicinity of the SED-sensors at some of the sites (see Table 1)". I think the use of pressure sensors should be mentioned in I.118-120 too.

Response: We thank the reviewer for this suggestion. We have included the following sentence in this part of the manuscript in Line 121: "The phase of the tidal level was

provided by accompany pressure sensors (for water level measurements) deployed close to SED sensors."

2.6 I.105-115: it would be nice to add some information here (or in Fig.2) regarding the size (i.e., total length) of the SED sensors. Response: This suggestion is very helpful, and we have revised the Figure 2 to indicate the size of the SED sensors.

2.7 DATA: in the "profile\_all\_Mar\_2020.xlsx" file, the coordinates (LAT, LON) for all sites are reported in decimal degrees except for sites 9 and 10. I believe it'd be better to keep the decimal degree format throughout the whole dataset. This should be an easy fix. Also, I've noticed that coordinates for site 10 are reported in red: is there a specific reason behind this color choice or is it just a misprint?

Response: We would like to thank the reviewer for this comment. We have revised the file to keep the decimal degree format throughout the whole dataset. The red coordinates of site 10 was indeed a misprint of the geodetic coordinates (unit: meter). To keep the uniform style of coordinates (LAT, LON in decimal degree format), we have deleted these red columns in the updated version. The revised file was uploaded as a zip file and will be updated in the main dataset.

2.8 FIGURE: Figure 1: the use of blue color to highlight (some) land cover makes the figure a bit confusing I think. It would be probably better to lose such a color-code in favor of a much clearer one with gray=land and white=sea.

Response: We thank the reviewer for this suggestion. We have revised the Figure using the suggested color scheme.

2.9 MINOR COMMENTS I.31: "instrument cost" -> "costly instruments" is perhaps better. I.33: "salt marsh sites"->"salt-marsh sites". I.34-35: "bed level changes"->"bed-level changes" (please add hyphen throughout the whole manuscript). I.36: D50->D\_50 (use subscript throughout the whole manuscript) I.36: "some sites", please specify how many. I.46: "with"-> "of" global importance. I.61: a semicolumn is missing after "(Bouma et al., 2016)". I.82: "the current"-> "such" I.84: lose "current" I.94: "in 2013-2017" -> "in the period 2013-2017"; also "from 9-20 months"->"from 9 to 20 months"

Response: We thank the reviewer for all these comments, and we have revised the manuscript accordingly.

Please also note the supplement to this comment: https://essd.copernicus.org/preprints/essd-2020-78/essd-2020-78-AC1supplement.zip

Fig. 2.

---

## Author Response (AR2)

Editor
Earth System Science Data

Dec 4, 2020

Dear Editor,

We would like to thank you for your rapid handling and response to our submission. Please find enclosed the revised manuscript for the *Earth System Science Data*, entitled "*Synchronized high-resolution bed-level change and biophysical data from ten marsh-mudflat sites in northwestern Europe*" [Paper # essd-2020-78], and detailed list of our point-by-point reply to the comments of the two reviewers. The line numbers of updated texts are marked according to the manuscript with changes tracked below.

We highly appreciated the comments made by the associate editor and reviewers, as they enabled us to improve our manuscript. In the attached letter below, we have detailed how their comments were used to revise the manuscript. If you have further suggestions for improvement, we will be willing to make additional modification.

Looking forward to your reply.

Yours sincerely,
On behalf of all co-authors,
Heng Wang

E-mail: wangheng3@mail.sysu.edu.cn

**"Synchronized high-resolution bed-level change and biophysical data from ten marsh-mudflat sites in northwestern Europe"** [Paper # essd-2020-78]

Zhan Hu, Pim W.J.M. Willemsen, Bas W. Borsje, Chen Wang, Heng Wang, Daphne van der Wal, Zhenchang Zhu, Bas Oteman, Vincent Vuik, Ben Evans, Iris Möller, Jean-Philippe Belliard, Alexander Van Braeckel, Stijn Temmerman, Tjeerd J. Bouma

**Response to reviewers' comments**

**1. Response to Edward Anthony (Referee 1):**

**1.1** This is an interesting and useful paper that opens up on bed-level changes and their drivers in tidal flats in NW Europe. The paper basically presents high-resolution bed level data obtained from SED sensors, an innovative method of measuring short-term substrate variations on tidal flats, coupled with biophysical characteristics, and shows the potential link with drivers, notably storms. My main concern revolves around the wave-height measurements (data description and table 1).

**Response:**

We would like to thank the positive comments from the reviewer. We hope that the revised version responds adequately to your concerns and we remain available for further improvement. The concern related to the wave-height measurements is addressed in detail in point 1.7 below.

**1.2** The term bed level needs to be hyphenated throughout when used in the adjectival form of 'bed-level changes' (this is done in Fig. 4c!). I would consider data as being in the plural form: 'data are' instead of 'data is', etc.

**Response:**

We thank the reviewer for these linguistic suggestions. The suggested revisions are done for the whole manuscript. Data and dataset are both considered in the plural form.

**1.3** Title: Awkward title. High-resolution bed-level change is linguistically rather strange as I do not see how bed level change can be of high or low resolution. Measurements or data are of high or low resolution but not change. I suggest 'Synchronized high-resolution bed-level and biophysical data from .....'

**Response:**

We thank this suggestion. The new title of the manuscript is: "Synchronized

high-resolution bed-level change and biophysical data from ten marsh-mudflat sites in northwestern Europe".

**1.4** Abstract: Line 73: deployment of labour; Line 76: delete 'rare' to read: providing an opportunity

**Response:**

We thank these suggestions from the reviewer and have revised these parts of the manuscript accordingly.

*"In light of these limitations, SED-sensors (Surface Elevation Dynamics sensors) have been developed to record daily bed-level dynamics with high accuracy, while reducing the unit cost and labour during deployment (Hu et al., 2015)"* (Line 88)

**1.5** Site description: Line 84: delete 'the' to read from northwestern Europe and use lettering when indicating the number of sites: seven sites, one site, etc. Site 1, 7 etc is OK as you are identifying your sites. Line 96: repetition of measuring. Maybe use: The coordinates of the monitoring stations as well as the bathymetry of the measuring transects...

**Response:**

We thank these suggestions for the reviewer and have revised these parts of the manuscript accordingly.

**1.6** Methods: I know this is not a study on long-term net sedimentation-erosion rates but how do you account for subsidence from the SED data? Line 143: water-level data were obtained from nearby tidal gauge stations... Line 144: Diatoms act as biostabilizers...

**Response:**

We thank the reviewer for this comment. Indeed, our study is not on long-term net sedimentation-erosion rates, but it is focused on short-term bed-level changes. We did not account for deep subsoil subsidence as over the short-term periods considered here (typically 10-15 months) values of subsidence are mostly in the order of less than 1 or a few mm for the study areas, and therefore mostly much less than values of vertical bed-level changes recorded by the SED sensors:

*"We note that the SED technique does not include effects of deep subsoil subsidence on bed-level changes. For a typical deployment period of the SED sensors (10-15 months), subsidence in the study areas is mainly related to glacial isostatic adjustment after the last ice age, with values in the order of less than 1 or a few mm over the considered time periods (Vink et al., 2007), and therefore mostly much less than values of vertical bed-level changes recorded by the SED sensors."* (Line 167)

The suggested revisions on the use of hyphen are adapted in the revised manuscript: 'water level data' to 'water-level data' and 'bio-stabilizers' to 'biostabilizers'.

**1.7** Data description: Line 173: I am highly surprised by mean Hs values of 0.1 cm and 0.008 cm? These are meaningless and tantamount to zero wave energy! How did you measure such values? A 1 mm-high wave!!! I would suggest you refrain from quoting these values (this also goes for the very low Hs values in Table 1), and focus more on the higher Hs values and their variations in Figure 3 which are more meaningful and pertinent to bed-level change.

**Response:**

We thank the reviewer for this comment. We apologize that the very low Hs values (0.1 cm and 0.008 cm) here were due to a mistake in the calculation. The corrected values should be 10.4 cm and 11.3 cm, respectively. We also checked the mean Hs values in the Table 1, which are correct. Similar values can be found in previous field studies at these sites (e.g. Callaghan et al., 2010; Hu et al., 2015). The very low mean Hs value (1 cm) at Site 3 Baarland in this table is because this site is sheltered by an offshore sandbar that blocks almost all the incident waves (see Figure 1). Additionally, we provide more details related to the data-processing method of Tucker and Pitt (2001) and the source of the processing routines:

*"Significant wave height (Hs) and peak wave period (Tp) were derived from the dynamic wave pressure signals. The attenuation of pressure signals with water depth was corrected using the standard calculations methods as described in Tucker and Pitt, (2001). The attenuation correction was only applied over the frequency range 0.05-0.4 Hz and the maximum correction factor was set as 5 to avoid over-amplification of high frequency signals (i.e. noise). A detailed description and source of the data-processing routines can be found at http://neumeier.perso.ch/matlab/waves.html."* (Line 204)

**1.8** Conclusions Line 22: Due to the lack of 'relevant' (instead of desired).

**Response:**

We thank the reviewer's suggestion. The text is revised accordingly.

**1.9** Table and figures: Table 1: Surprised at the significant wave height data values! Figure 1: Please add geographical coordinates and scale. Figure 2: Please add a scale in (b) to show length of SED probe. Figure 3. Use black bar to indicate wave height in figure legend.

**Response:**

We thank reviewer's comments on the table and figures. The issue with significant wave height is addressed in point 1.7. All the suggestions with the figures are adapted in the revised manuscript. Please see the updated figures below.

**2. Response to Alvise Finotello (Referee 2)**

**2.1** The article is clear and well structured. Figures and tables are of good quality and appropriate to support the text. The dataset presented is most certainly new (even though some subsets of these data were already employed for individual studies on specific study-case), and I believe it has a strong potential for being useful in the future (also because the authors intend to further expand the dataset by adding new study cases and data).

Methods and material are described in sufficient detail (with some minor exceptions that I've listed here below), and the cited literature is appropriate in my view.

I encountered no issues in accessing the dataset, which I found is complete and well organized. I've only highlighted in my comments one minor suggestion regarding the format of geographic coordinates to improve data consistency throughout the dataset. Data are easy to read and ready-to-use.

**Response:**

We would like to thank the positive comments from the reviewer. We hope that the revised version responds adequately to your concerns and we remain available for further improvement.

**2.2** However, I share the concern of Referee 1 (see RC1 comments n.5 and n.7) regarding the reported low values of significant wave height (Hs) equal to 0.1 and 0.08 cm (text line 173). The authors refer to the work of Tucker and Pitt (2001) for the calculation of Hs and wave period from dynamic wave pressure signals. I believe the text should be briefly expanded to provide an explanation regarding this procedure, which are the potential sources of error, and what is the physical limit – if any – below which the derived Hs values should be disregarded due to (e.g.) instrument accuracy.

**Response:**

We thank the reviewer for this comment. This response was shared with that of the comment 1.7. We apologize that the very low Hs values (0.1 cm and 0.008 cm) were due to a mistake in the calculation. The corrected values should be 10.4 cm and 11.3 cm, respectively. We also checked the mean Hs values in the Table 1, which are correct. Similar values can be found in previous field studies at these sites (e.g. Callaghan et al., 2010; Hu et al., 2015). The very low mean Hs value (1 cm) at Site 3 Baarland is because this site is sheltered by an offshore sandbar that blocks almost all the incident waves (see Figure 1). Additionally, we provide more details related to the data-processing method of Tucker and Pitt (2001) and the source of the processing routines:

  *"Significant wave height (Hs) and peak wave period (Tp) were derived from the dynamic wave pressure signals. The attenuation of pressure signals with water depth was corrected using the standard calculations methods as described in Tucker and Pitt, (2001). The attenuation correction was only applied over the frequency range 0.05-0.4 Hz and the maximum correction factor was set as 5 to avoid over-amplification of high frequency signals (i.e. noise). A detailed description and source of the data-processing routines can be found at http://neumeier.perso.ch/matlab/waves.html."* (Line 204)

**2.3** l.1: the wording chosen for the title is a bit strange, because it seems that "high resolution" is referred to bed level (hyphenation is missing, by the way). I would suggest changing it to "High-resolution synchronized data of bed-level changes and biophysical data from…", or something similar. Reading the abstract, it becomes clear that the focus of the paper is also (if not mostly) on salt marshes, so I find the choice of using "tidal flats" in the title a bit odd. My impression is that making explicitly clear that the paper aims to investigate the dynamics of both tidal flats and marshes would be much better.

**Response:**

We would like to thank the reviewer for this constructive comment. Now the title is revised as: "Synchronized high-resolution bed-level change and biophysical data from ten marsh-mudflat sites in northwestern Europe"

**2.4** l.53-54: "The evaluation of...", there's something strange in this sentence. Perhaps "Evaluating the impact of changing sea level and increasing storminess on these valuable coastal ecosystems is of socioeconomic importance" is more clear.

**Response:**

We agree with the reviewer's comment, and the text is revised accordingly.

**2.5** l.118-120: how was the low-tide defined/measured? Directly by the SED sensor (if so, how) or rather by coupling SEDs with a pressure gauge to measure water elevation? Later in the text (l.133) it is specified that "we deployed pressure sensors 0.05 m 0.10 m above the bed in the vicinity of the SED-sensors at some of the sites (see Table 1)". I think the use of pressure sensors should be mentioned in l.118-120 too.

**Response:**

We thank the reviewer for this suggestion. We have included the following sentence in this part of the manuscript:

*"The tidal fluctuation of water level was recorded by pressure sensors deployed close to SED sensors."* (Line 176)

**2.6** l.105-115: it would be nice to add some information here (or in Fig.2) regarding the size (i.e., total length) of the SED sensors.

**Response:**

This suggestion is very helpful, and we have revised the Figure 2 to indicate the size of the SED sensors.

**2.7** DATA: in the "profile_all_Mar_2020.xlsx" file, the coordinates (LAT, LON) for all sites are reported in decimal degrees except for sites 9 and 10. I believe it'd be better to keep the decimal degree format throughout the whole dataset. This should be an easy fix. Also, I've noticed that coordinates for site 10 are reported in red: is there a specific reason behind this color choice or is it just a misprint?

**Response:**

We would like to thank the reviewer for this comment. We have revised the file to keep the decimal degree format throughout the whole dataset. The red coordinates of site 10 was indeed a misprint of the geodetic coordinates (unit: meter). To keep the uniform style of coordinates (LAT, LON in decimal degree format), we have deleted

these red columns in the updated version. The revised file has been uploaded to together with the main dataset.

**2.8** FIGURE: Figure 1: the use of blue color to highlight (some) land cover makes the figure a bit confusing I think. It would be probably better to lose such a color-code in favor of a much clearer one with gray=land and white=sea.

**Response:**

We thank the reviewer for this suggestion. We have revised the Figure using the suggested color scheme.

**2.9** MINOR COMMENTS

l.31: "instrument cost" –> "costly instruments" is perhaps better.

l.33: "salt marsh sites"->"salt-marsh sites".

l.34-35: "bed level changes"->"bed-level changes" (please add hyphen throughout the whole manuscript).

l.36: D50->D_50 (use subscript throughout the whole manuscript)

l.36: "some sites", please specify how many.

l.46: "with"-> "of" global importance.

l.61: a semicolumn is missing after "(Bouma et al., 2016)".

l.82: "the current"-> "such"

l.84: lose "current"

l.94: "in 2013-2017" -> "in the period 2013-2017"; also "from 9-20 months"->"from 9 to 20 months"

**Response:**

We thank the reviewer for all these comments, and we have revised the manuscript accordingly.

[revised manuscript text omitted]